# An Evaluation System for Assessing the Operational Efficiency of Urban Combined Sewer Systems Using AHP—Fuzzy Comprehensive Evaluation: A Case Study in Shanghai, China

Hongwu Wang [1,*], Ming Yan [1], Yuan Gao [2], Yanqiong Wang [1] and Xiaohu Dai [1]

[1] Key Laboratory of Urban Water Supply, Water Saving and Water Environment Governance in the Yangtze River Delta of Ministry of Water Resources, National Engineering Research Center for Urban Pollution Control, State Key Laboratory of Pollution Control and Resource Reuse, College of Environmental Science and Engineering, Tongji University, Shanghai 200092, China

[2] Shanghai Urban Construction Design and Research Institute (Group) Co., Ltd., Shanghai 200125, China

[*] Correspondence: wanghongwu@tongji.edu.cn

**Abstract:** In recent years, China's urban smart water management has focused on enhancing the quality, efficiency, energy conservation, flood prevention, and control of combined sewer overflow (CSO) and urban waterlogging. To evaluate the combined sewer system's operation efficiency effectively, a comprehensive evaluation system was established. This system incorporated expert scoring and the coefficient of variation method, considering 31 specific indexes to assess six key aspects: CSO control, waterlogging control, stable wastewater transportation, pipeline management and maintenance, energy conservation, and smart water affairs. The AHP—Fuzzy comprehensive evaluation was employed for evaluation, combining AHP for index weighting and fuzzy comprehensive evaluation for quantitative scoring. The system was applied to assess the operation efficiency of a specific area in Shanghai from 2020 to 2022. Results showed progress in CSO control and energy conservation, with operation efficiency improving from low in 2020 to moderate in 2021 and good in 2022. However, waterlogging control and pipeline management still require improvement in the combined sewer area. Overall, the evaluation system provides valuable insights into the system's performance, identifying areas for targeted enhancement and emphasizing the need for further improvements to achieve optimal operation efficiency.

**Keywords:** combined sewer system; evaluation of operation efficiency; analytic hierarchy process; fuzzy comprehensive evaluation; comprehensive evaluation method

## 1. Introduction

Evaluating the operational efficiency of urban sewer systems holds immense significance, as it guides their development direction, informs multi-objective decision making, and aids in system optimization. Such evaluations enable us to identify system issues, assess the current state, predict various risks, and provide valuable insights for the development, decision making, and optimization of drainage systems. In China, the rise of new technologies, such as artificial intelligence, digital twins, and the Internet of Things, has catalyzed the rapid development of 'smart water affairs' [1]. Guided by principles like 'low-carbon', 'green', 'energy-saving', 'scientific', and 'harmonious', several 'key technologies for intelligent urban sewer system treatment' have emerged. Within this context, evaluating the operational efficiency of sewer systems has become a crucial focus. This paper delves into the evaluation of operation efficiency for combined sewer systems.

Practical research on the evaluation of sewer system operational efficiency encompasses various aspects. Researchers such as Chen F [2], Leimgruber J et al. [3], and Zhang D et al. [4] have examined combined sewer overflow control and waterlogging management.

Zheng M et al. [5], Okwori E et al. [6], and Ghavami S et al. [7] have assessed the performance of drainage networks. Ananda J et al. [8] have devised performance evaluation systems for sewer management, while Baah K [9], Ba Z et al. [10], and Liu W et al. [11] have investigated safety risk assessments of sewer systems. Additionally, Nam S [12] and Wang J et al. [13] have conducted comprehensive evaluations of drainage systems.

As evident from the above, comprehensively evaluating sewer system operational efficiency involves multiple facets and angles. This multi-dimensional challenge is often referred to as 'multi-criteria evaluation', and it employs various methods, as outlined in Table 1.

**Table 1.** Typical multi-criteria evaluation method.

| Method | Features | Application Cases |
|---|---|---|
| AHP | Subjective weighting method; suitable for consulting experienced experts, convenient and efficient, but too subjective; applicable to index systems with both quantitative and qualitative indexes | Evaluate the performance of the operation of the Korean sewer system [12] |
| AHP-FCE | Subjective weighting method; qualitative indexes can be quantified through membership functions | Comprehensive evaluation of the state and operation efficiency of the drainage network in a region of Huai'an City, China [13] |
| AHP-DEA | Subjective weighting method; suitable for multi-criteria evaluation and subsequent optimization; significantly improving the optimality and fairness of the evaluation system | Combining Bayesian Networks to evaluate the probability of failure and consequences in sewer pipelines [7] |
| AHP-GRA | Subjective weighting method; applicable to problems where the sample data are relatively small and regression analysis cannot be conducted; quantifying scores through grey correlation analysis | Risk assessment of combined sewer pipes [14] |
| FAHP | Subjective weighting method; reduced the difficulty of determining the consistency of the matrix, which is a fuzzification of AHP, resulting in increased uncertainty | Research the allocation of the flood drainage rights about each administrative region [4] |
| ANP-FCE | Subjective weighting method; allow mutual control and influence between indexes; suitable for complex systems | Risk assessment of the industrial park drainage system [10] |
| DEA | Objective weighting method; a planning model established from the perspective of input–output can be used for multi-objective optimization and decision making | Measuring the economic efficiency of Australian wastewater treatment services [8]; Using tobit regression analysis to evaluate the efficiency of wastewater company [15] |
| Entropy | Objective weighting method; determine indicator weights based on information entropy theory; applicable to quantitative indexes | |
| Entropy-TOPSIS | Objective weighting method; applicable to ranking or comparison problems of multi-objective evaluations | Assessment of waterlogging control capacity in 31 provinces in China [16] |
| Entropy-FCE | Objective weighting method; qualitative indexes can be quantified through membership functions | Urban waterlogging risk assessment [17] |
| ANN-FCE | Objective weighting method, with adaptive learning ability; suitable for situations with a large amount of high-quality data available | Comprehensive assessment of urban waterlogging control capabilities and risks [18]; Using the PSO-ELM model to predict and diagnose drainage pipelines [5] |
| AHP-Entropy | Subjective and objective combination weighting method; this not only reduces the subjectivity of AHP, but also avoids the sensitivity of entropy weight method to data, and uses membership function to quantify the evaluation results | Performance evaluation of rainwater concrete pipes [19]; The operation efficiency of a regional sewer pipe network in a city was evaluated [11] |
| AHP-ANN | Subjective and objective combination weighting method; suitable for situations with a large amount of high-quality data available | Establish of an evaluation model of GA-optimized BP neural network to evaluate the health status of sewer pipes [20] |
| Fuzzy Borda | Subjective and objective combination weighting method; combining multiple evaluation methods to improve the reliability and applicability of evaluation | Combining entropy, TOPSIS, efficiency coefficient, FCE, etc. through Borda combination evaluation method to evaluate the vulnerability of rainwater pipe network [21] |

The primary methods employed in the multi-criteria evaluation approach are the Analytic Hierarchy Process (AHP) and the Entropy Weight Method. AHP is a subjective method for determining weights, relying on expert judgments. It finds its suitability in scenarios where both qualitative and quantitative indicators coexist. In contrast, the Entropy Weight Method is an objective technique that derives weights from indicator information entropy. It is particularly well suited for quantitative indices. In addressing issues related to evaluation uncertainty, subjectivity, and the quantification of qualitative indices, several additional methods are employed. These include fuzzy mathematics theory, gray correlation analysis (GRA), artificial neural networks (ANNs), data envelopment analysis (DEA), and the Technique for Order Preference by Similarity to an Ideal Solution (TOPSIS). When these methods are used in combination, it is often referred to as a comprehensive evaluation approach.

In this study, the index system contains numerous elements necessitating qualitative evaluation, making the AHP a fitting choice. Moreover, owing to the presence of several qualitative indicators, quantifying the evaluation results becomes a challenging task. To address this challenge, we have introduced the principles of fuzzy mathematics into our evaluation framework. It is noteworthy that the integration of AHP and fuzzy mathematics typically can be divided into two types: one is Fuzzy Analytic Hierarchy Process (FAHP) and the other is AHP–Fuzzy Comprehensive Evaluation (FCE), which are depicted in Figure 1. FAHP involves fuzzifying the assigned weights, while Fuzzy Comprehensive Evaluation pertains to the fuzzification of the evaluation results. Due to the authoritative experts in this field who were consulted to assign the weights in this study, fuzzy mathematics was only used for the evaluation results. As such, this study leans towards an AHP-FCE approach (Figure 1b) to establish a set of evaluation index systems for assessing the operation efficiency of an urban combined sewer system.

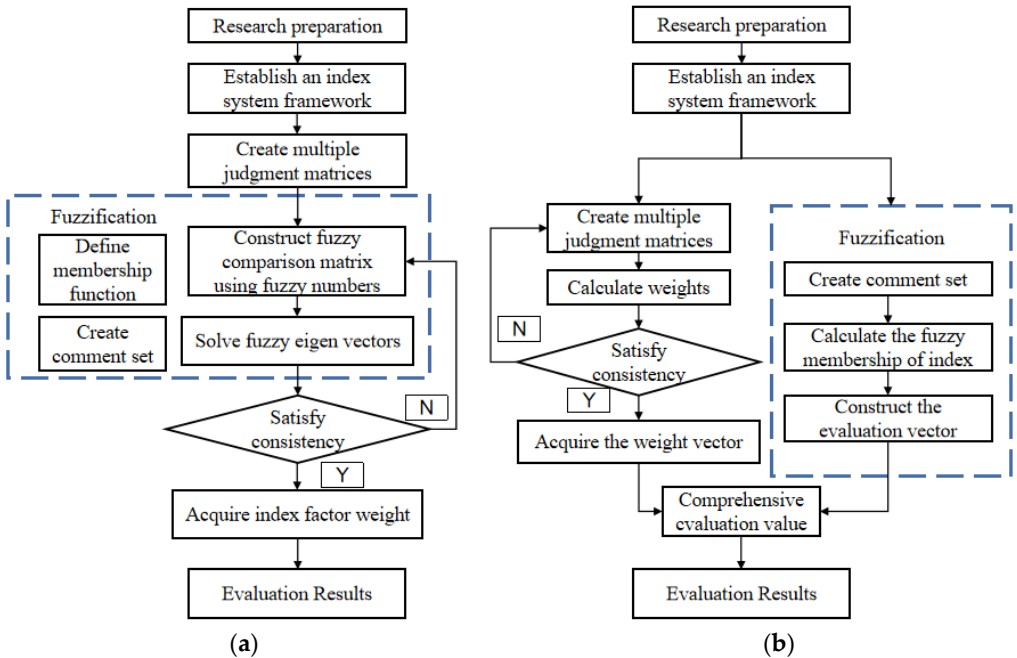

**Figure 1.** Flow chart of the evaluation method combining analytic hierarchy process and fuzzy mathematics. Panel (**a**) shows the flow chart of FAHP evaluation; panel (**b**) shows the flow chart of AHP-fuzzy comprehensive evaluation.

## 2. Evaluation System for Operation Efficiency of Combined Sewer System

### 2.1. Establishment of the Evaluation Index System

The operational efficiency of a combined sewer system encompasses various facets, necessitating the development of a comprehensive multi-objective evaluation system. This

system is designed to provide a precise, objective, and scientifically grounded assessment of both the system's overall performance and its individual components. The ultimate goal is to offer invaluable guidance for the development and construction of efficient drainage systems. In this study, we conducted several expert assessments and thoughtfully selected evaluation metrics after carefully analyzing the real-world conditions of the combined sewer system. These metrics underwent rigorous screening, considering six critical aspects: combined sewer overflow (CSO) control, waterlogging management, seamless wastewater transport, pipeline maintenance, energy efficiency, and smart water management. By incorporating a wide range of perspectives and aspects into the evaluation process, our multi-objective comprehensive evaluation system aims to provide a comprehensive and precise assessment of the operational efficiency of the combined sewer system. The insights gained from this assessment will serve as a valuable reference, offering essential guidance for optimizing and enhancing the overall performance of the sewer system.

Throughout the index selection process, we adhered to principles of scientific rigor, completeness, principal components, and independence. This process involved various methodologies, including literature frequency analysis, expert consultations, and theoretical analysis.

The primary sources of reference for conducting literature frequency analysis are categorized into two main sections. The first comprises indexes that have been utilized in previous studies published in research papers. The second section encompasses indexes recommended for use by international organizations. In line with relevant studies on the assessment of sewer system operational efficiency, these can be broadly classified into four categories based on their evaluation objectives: (1) combined sewer overflow control and waterlogging management, (2) prediction and diagnosis of pipe network performance, (3) sewer system management and operation, and (4) comprehensive sewer system evaluation. A compilation of common indexes employed in these studies is presented in Table 2.

Several international organizations, including the International Water Association (IWA) [22], the International Benchmarking Networking of Water Supply and Drainage Performance (IBNET) [23], the Water Services Regulation Authority (OFWAT) [24], the U.S. Environmental Protection Agency (US EPA) [25], and the American Water Works Association (AWWA) [26], have introduced diverse evaluation metrics, as detailed in Table 3. These metrics comprehensively address various aspects of effectiveness evaluation in drainage system operation and planning. Based on the characteristics of these indicators, they can be categorized into four distinct types: ecological effects, social effects, economic effects, and network planning.

**Table 2.** Summary of classification of sewer system operation efficiency evaluation objectives.

| Evaluation Objectives | Common Indexes | Authors |
|---|---|---|
| Combined sewer overflow control and waterlogging control | **Descriptive indexes:** waterlogging nodes, waterlogging volume, waterlogging duration, waterlogging area, waterlogging depth, overflow water volume, overflow duration, overflow distribution, etc.; **Predictive indexes:** proportion of waterlogged pipe sections, potential waterlogging hazards, proportion of overflow volume, potential overflow pollution, etc.; **Guiding indexes:** waterlogging reduction potential, overflow reduction potential, regulation and storage capacity, dispatching capacity, etc. | Chen F(2016) [2], Leimgruber J et al.(2018) [3], Jiang Z(2020) [16], Zhang D et al.(2020) [4], Cai Z et al.(2020) [18] |
| Prediction and diagnosis of pipe network performance | **Pipeline network design indexes:** drainage pipe diameter, storm water outlet elevation, dispatching capacity, storage capacity, etc.; **Hydraulic performance indexes:** pipeline water depth, pipeline flow velocity, pipeline fullness, pipeline bearing capacity, etc.; **Pipeline status indexes:** pipe age, pipe material, burial depth, cushion, pipeline blockage, pipeline inspection, pipeline maintenance, etc. | Zheng M et al.(2020) [5], Okwori E et al.(2020) [6], Ghavami S et al.(2020) [7], Yang L et al.(2021) [20], Jin H et al.(2021) [19], Wang Z et al.(2018) [17], He F et al.(2023) [21] |

**Table 2.** *Cont.*

| Evaluation Objectives | Common Indexes | Authors |
|---|---|---|
| Management and operation of sewer systems | **Ecological effect indexes:** greenhouse gas emissions, drainage pollution, environmental benefits, etc.; **Social effect indexes:** service population, customer service satisfaction, etc.; **Economic effect indexes:** maintenance costs, investment in fixed assets, etc.; | Young J (2017) [15], Lee J et al. (2018) [27], Ananda J (2020) [8] |
| Comprehensive evaluation of sewer systems | **Pipe network planning indexes:** pipe network coverage, pipe network density, pipe network structure at all levels, etc.; Sewage treatment capacity, drainage pipe network density, sewage treatment rate, low carbon conservation, recycling, natural symbiosis, value creation, smart management and control, etc. | Wang Z et al. (2018) [17], Nam S et al.(2019) [12], Wang J et al.(2022) [13] |

**Table 3.** Operation efficiency of sewer system indexes published by international organizations.

| IWA | IBNET | OFWAT | US EPA | AWWA |
|---|---|---|---|---|
| Water Resources | Service Coverage | Sewer overflow | Serving population | Pipe service life |
| Personnel Index | Water Consumption and Production | Applicable drainage non-infrastructure | Total length of drainage pipe | Wastewater collection cost per km of pipeline |
| Operating Index | Treated wastewater volume | Applicable drainage infrastructure | Pipe internal and external status | Customer service satisfaction |
| Fixed Assets | Pipe network performance | Leakage | Pipe slope | Operation and maintenance cost per km of pipeline |
| Service Quality | Cost and personnel | GHG emission | Pipe length, diameter, material, etc. | Number of wastewater treatment facilities |
| Financial Situation | Service quality | Serious pollution incident | Construction status of ancillary sewer facilities | Treated wastewater volume |
| | Fixed asset investment | Sewage Discharge Permit Compliance | Use of auxiliary drainage facilities | Sewage treatment rate |
| | Revenue and expenditure Financial performance | Economic index | | Income and expenditure status Pipe network coverage |

Drawing from the referenced drainage system indicators and considering the specific conditions of Shanghai's combined sewer network, we initially identified a total of forty-two indexes, spanning six critical dimensions. Detailed definitions and the significance of these indexes can be found in Table 4.

**Table 4.** Explanation of evaluation indexes for operation efficiency of combined sewer system.

| Rule Layer | Index Layer | Index Meaning |
|---|---|---|
| CSO control | Pumping station discharge volume ($m^3$) | The discharge volume of trunk line pumping stations discharged into rivers in rainy days. |
| | Pumping station discharge frequency (%) | The frequency of discharge of trunk line pumping stations into the river within a year. |
| | Pumping station discharge time (min) | The average time for the trunk line pumping station to discharge the river. |
| | Wastewater treatment plant overflow ratio (%) | The ratio of the overflow of the wastewater treatment plant to the water inflow at the end of the pipe network. |
| | Wastewater treatment plant overflow frequency (%) | Overflow frequency of wastewater treatment plant in one year. |
| | Wastewater treatment plant overflow time (min) | Average duration of wastewater treatment plant overflow. |
| | Overflow potential hazard | Hazardous consequences of overflow (Pumping station discharge to the river + overflow in wastewater treatment plant). |
| | Overflow reduction potential | Construction level of overflow pollution control facilities. |
| | Area of influence of overflow | Area affected by pollution from overflow (Pumping station discharge to the river + overflow in wastewater treatment plant). |
| | Interception capacity | Overall interception capacity of trunk line pumping station. |

Table 4. *Cont.*

| Rule Layer | Index Layer | Index Meaning |
|---|---|---|
| Waterlogging control | Ponding depth (m) | The maximum rainwater depth at which ponding occurs within the catchment area. |
| | Ponding frequency (%) | Frequency of ponding during rainfall. |
| | Ponding time ratio | The ratio of the maximum duration of ponding to the rainfall duration. |
| | Ponding area ratio (%) | Percentage of catchment areas that are flooded. |
| | Ponding point ratio (%) | The ratio of points with ponding water within one year to the total monitoring points. |
| | Fullness Degree | Average fullness degree of downstream trunk pipelines during rainfall. |
| | Overloaded pipeline ratio (%) | The ratio of pipelines overloaded during rainfall to total pipelines. |
| | Waterlogging reduction potential | Construction level of waterlogging facilities. |
| | Waterlogging potential hazards | Harmful consequences of waterlogging. |
| Stable transportation of wastewater | Variation coefficient of inflow volume | Changes in wastewater volume at the end of the drainage pipeline network. |
| | Variation coefficient of inflow quality | Changes in wastewater quality at the end of the drainage pipeline network. |
| | Storage capacity | The storage capacity of the drainage system. |
| | Pumping station dispatch capability | Dispatching capability of drainage system trunk line pumping station. |
| | Forebay water level of outlet pump room (m) | Forebay wastewater level of outlet pump room at end of pipeline network. |
| Management and maintenance of pipeline | Pipeline network length per capita (km/10,000 people) | The length of the pipeline network per 10,000 people in the drainage pipe network service area. |
| | Urban pipeline network investment (10,000 yuan/km$^2$) | Urban drainage investment divided by urban built-up area. |
| | Wastewater delivery capacity (%) | The water delivery of the pipeline network divided by the design flow of the wastewater treatment plant. |
| | Maintenance and update cost (10,000 yuan/km) | Equipment maintenance and renewal costs per kilometer of drainage pipeline network facilities. |
| | Pipeline network density (km/km$^2$) | Drainage pipe network construction density. |
| | Wastewater treatment rate (%) | Matching degree of wastewater supply in the service area and terminal treatment capacity. |
| | Operator level | Working years and professional titles of sewer system operators. |
| | Pipeline inspection (%) | Percentage of lengths of sewer pipes inspected annually. |
| | Amount of sludge removed (t/km) | Amount of pipeline sludge removed per kilometer of pipeline. |
| Energy conservation | Total energy consumption (kW·h) | The total energy consumption of sewer trunk line transportation, flood control, etc. |
| | Wastewater transportation unit consumption (kW·h/1000 m$^3$) | Power consumption per thousand cubic meters of water transported by the drainage pipe network. |
| | Environmental benefits | Comprehensive benefits to the urban environment. |
| | Increase energy consumption value | Total energy consumption (tons of standard coal) divided by industrial added value (10,000 yuan). |
| Smart water affairs | Inflow monitoring ratio (%) | Proportion of liquid level monitoring installed in the inflow section of the main branch line on the trunk sewer line. |
| | Layout of pipeline monitoring points (points/10 km) | On-line monitoring points in every kilometer of pipeline. |
| | Intelligent decision | Intelligent decision-making level in drainage network diagnosis. |
| | Intelligent control | Sewage plant–drainage network–river joint dispatch and intelligent control level. |
| | Digitalization degree of pipe network (%) | Proportion of digitally managed pipe segments. |

### 2.2. Optimization of Evaluation Indexes

The coefficient of variation (CV) is a useful metric for comparing data dispersion among multiple datasets. It becomes particularly valuable when dealing with datasets that have vastly different measurement scales or dimensions. Directly comparing standard deviations in such cases may not be appropriate, as it requires eliminating measurement and dimensional influences. The CV, in contrast, is the ratio of the standard deviation to the mean of the original data. This makes the coefficient of variation dimensionless, allowing for objective comparisons among multiple datasets.

In this study, the evaluation indexes were optimized using a combination of the questionnaire scoring method and the coefficient of variation method. Initially, the 42 indexes were reassessed to identify representative, highly important, and reliable evaluation indicators. Due to differences in indicator dimensions, the coefficient of variation was employed to analyze the scoring results.

First, we designed a questionnaire to gauge the importance of the initial 42 indexes. The questionnaire was distributed to experts and personnel involved in various aspects of the drainage system, who provided ratings for the significance of these primary indicators. We collected a total of 49 valid questionnaires. Next, we analyzed the scoring results from the 49 experts to create an expert evaluation quantification table. For each index, we calculated the average score and variation coefficient. Based on these values, we selected evaluation indicators with high average scores and small coefficients of variation for subsequent quantitative or qualitative evaluation.

To ensure the reliability of the survey results, we introduced an expert-level coefficient to adjust the scores, considering factors such as work positions, professional titles, and years of experience of the 49 scoring experts. The correction coefficient values are provided in Table 5, and the composition of the expert group is illustrated in Figure 2.

**Table 5.** The value of the correction coefficient of the expert group for rescreen of evaluation indexes.

| Work Position | $k_1$ | Professional Title | $k_2$ | Working Years | $k_3$ |
|---|---|---|---|---|---|
| Research | 1 | Senior | 1 | Over 10 years | 1 |
| Design | 0.9 | Deputy Senior | 0.9 | 5–10 years | 0.8 |
| Management | 0.7 | Intermediate | 0.7 | 3–15 years | 0.7 |
| Operation | 0.5 | Primary | 0.5 | Less than 3 years | 0.5 |

Note: $k_1$ is the correction coefficient for different work positions; $k_2$ is the correction coefficient for different professional titles; $k_3$ is the correction coefficient for different working years.

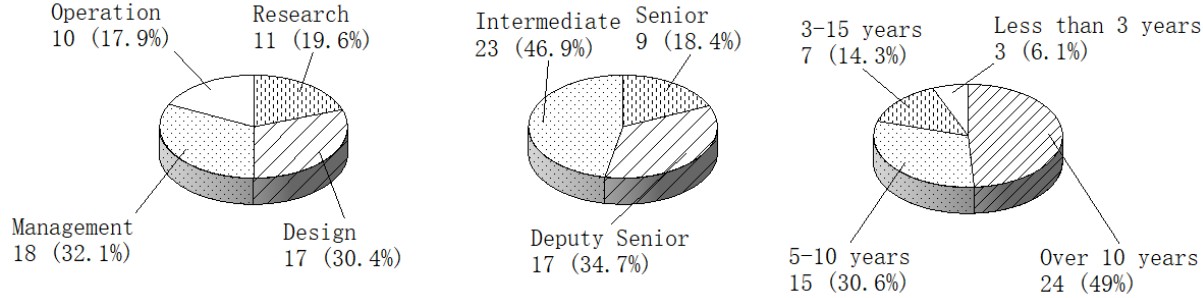

(**a**) the component of work position    (**b**) the component of professional title    (**c**) the component of working years

**Figure 2.** Component of the members of the expert group for rescreen of evaluation indexes of the combined sewer system. Panel (**a**) shows the component of work position of the advisory expert group; panel (**b**) shows the component of professional title of the advisory expert group; panel (**c**) shows the component of working years of the advisory expert group.

The scoring experts represented institutions including the Shanghai Municipal Design Institute, Shanghai Drainage Department, Shanghai Water Group, Shanghai Urban Construction Design Institute, and Tongji University.

The index screening process is outlined in Table 6. Indexes with high mean importance values were retained and low coefficients of variation, while removing those with low mean importance values and low coefficients of variation. Indexes with significant coefficients of variation were subject to further discussion.

**Table 6.** Description of the method of screening indexes using the coefficient of variation method.

| Mean | CV | Situation Description | Process Result |
|------|-----|-----------------------|----------------|
| ≥4 | ≤0.3 | A relatively large number of experts believe that the importance of this index is high | Adopt |
| ≥4 | >0.3 | The importance of this index is relatively high, but there are large differences in the scores among experts | Further argumentation, prudently adopt |
| <4 | ≤0.3 | A relatively large number of experts believe that the importance of this index is low | Eliminate |
| <4 | >0.3 | The importance of this index is relatively low, but there are large differences in the scores among experts | Further argumentation, prudently eliminate |

Note: CV is the abbreviation for coefficient of variation.

Following a comprehensive analysis and summarization of the experts' scoring results, we have compiled and presented the expert group's scoring outcomes in Table 7. In this table, the average score reflects the collective judgment of the experts regarding the importance of each specific index. A higher average score signifies greater importance, aligning with the consensus view of the experts. Conversely, the coefficient of variation indicates the degree of divergence among the experts in their assessments of the importance of each index. A higher coefficient of variation score indicates a greater level of disparity among the experts regarding the significance of a particular index.

After applying the Analytic Hierarchy Process (AHP) to weight the indexes, it became evident that the weight assigned to certain indexes was disproportionately small. To maintain the consistency of the judgment matrix and ensure the representativeness of the selected indexes, we decided to exclude three specific indexes: interception capacity, amount of sludge removed, and operator level.

Six indexes were removed after extensive discussions among experts for the following reasons:

(1) Pumping station discharge time: Despite some experts believing it could effectively represent overflow duration, it was removed due to the challenges in accurately measuring pumping station discharge into the river, especially during rainy days.

(2) Wastewater treatment plant overflow time: Similar to the previous index, it was removed because accurately measuring wastewater treatment plant discharge into the river, especially during rainy days, posed significant challenges.

(3) Ponding area ratio: Although some experts saw potential in this indicator for characterizing waterlogging severity, it was removed because, in practice, ponding areas can result from topographical factors unrelated to waterlogging.

(4) Fullness degree: While some experts thought this index could characterize hydraulic properties within drainage pipes and aid in predicting flooding, others believed it was mostly reflective of full pipe flow during rainy days, rendering it irrelevant for evaluation. It was removed based on the study's actual context.

(5) Waterlogging potential hazards: This index was challenging to measure accurately, leading to its removal.

(6) Variation coefficient of inflow quality: Although considered by some to reflect overflow pollution extent and hazards, experts argued that changes in this index were not directly related to overflows and waterlogging. As a result, this indicator was deleted based on expert recommendations.

**Table 7.** Index rescreening expert group scoring results.

| Rule Layer | Index Layer | Mean | CV | Result |
|---|---|---|---|---|
| CSO control | Pumping station discharge volume | 4.37 | 0.2611 | Adopted |
| | Pumping station discharge frequency | 3.97 | 0.3191 | Adopted after expert argumentation |
| | Pumping station discharge time | 3.63 | 0.3847 | Eliminated after expert argumentation |
| | Wastewater treatment plant overflow ratio | 4.17 | 0.2809 | Adopted |
| | Wastewater treatment plant overflow frequency | 3.98 | 0.3243 | Adopted after expert argumentation |
| | Wastewater treatment plant overflow time | 3.74 | 0.3532 | Eliminated after expert argumentation |
| | Overflow potential hazard | 4.07 | 0.2647 | Adopted |
| | Overflow reduction potential | 3.89 | 0.2754 | Eliminated |
| | Area of influence of overflow | 4.14 | 0.2418 | Adopted |
| | Interception capacity | 4.37 | 0.2355 | Eliminated due to small weight |
| Waterlogging control | Ponding depth | 4.57 | 0.2366 | Adopted |
| | Ponding frequency | 4.21 | 0.2678 | Adopted |
| | Ponding time ratio | 4.37 | 0.2677 | Adopted |
| | Ponding area ratio | 3.82 | 0.3360 | Eliminated after expert argumentation |
| | Ponding point ratio | 4.14 | 0.2740 | Adopted |
| | Fullness Degree | 3.79 | 0.3569 | Eliminated after expert argumentation |
| | Overloaded pipeline ratio | 3.94 | 0.2865 | Adopted after expert argumentation |
| | Waterlogging reduction potential | 4.08 | 0.2833 | Adopted |
| | Waterlogging potential hazards | 3.73 | 0.3182 | Eliminated after expert argumentation |
| Stable transportation of wastewater | Variation coefficient of inflow volume | 4.32 | 0.2523 | Adopted |
| | Variation coefficient of inflow quality | 3.89 | 0.3191 | Eliminated after expert argumentation |
| | Storage capacity | 4.19 | 0.2317 | Adopted |
| | Pumping station dispatch capability | 4.20 | 0.2390 | Adopted |
| | Forebay water level of outlet pump room | 3.96 | 0.2827 | Adopted after expert argumentation |
| Management and maintenance of pipeline | Pipeline network length per capita | 4.27 | 0.2431 | Adopted |
| | Urban pipeline network investment | 4.16 | 0.2453 | Adopted |
| | Wastewater delivery capacity | 4.60 | 0.2109 | Adopted |
| | Maintenance and update cost | 4.19 | 0.2553 | Adopted |
| | Pipeline network density | 4.40 | 0.2127 | Adopted |
| | Wastewater treatment rate | 4.27 | 0.2426 | Adopted |
| | Operator level | 4.21 | 0.2498 | Eliminated due to small weight |
| | Pipeline inspection | 4.00 | 0.2387 | Adopted |
| | Amount of sludge removed | 4.37 | 0.2154 | Eliminated due to small weight |
| Energy conservation | Total energy consumption | 4.21 | 0.2678 | Adopted |
| | Wastewater transportation unit consumption | 3.93 | 0.2845 | Adopted after expert argumentation |
| | Environmental benefits | 4.10 | 0.2401 | Adopted |
| | Increase energy consumption value | 3.86 | 0.2826 | Eliminated |
| Smart water affairs | Inflow monitoring ratio | 4.23 | 0.2350 | Adopted |
| | Layout of pipeline monitoring points | 4.16 | 0.2345 | Adopted |
| | Intelligent decision | 4.24 | 0.2272 | Adopted |
| | Intelligent control | 4.07 | 0.2527 | Adopted |
| | Digitalization degree of pipe network | 4.10 | 0.2448 | Adopted |

Following two rounds of screening, encompassing the initial index selection and subsequent re-evaluation, we arrived at a final set of 31 specific indexes. These indexes

comprehensively cover six critical dimensions: CSO control, waterlogging control, stable wastewater transportation, pipeline management and maintenance, energy conservation, and smart water affairs.

Following two rounds of screening, namely, the primary screening of indexes and the rescreening of indexes, a final set of 31 specific indexes were obtained, covering six crucial aspects: CSO control, waterlogging control, stable transportation of wastewater, pipeline management and maintenance, energy conservation, and smart water affairs.

## 3. AHP—Fuzzy Comprehensive Evaluation Method

The AHP—Fuzzy Comprehensive Evaluation method, also known as the AHP-FCE method, combines the Analytic Hierarchy Process (AHP) with Fuzzy Comprehensive Evaluation (FCE). This approach aims to capitalize on the strengths of both AHP and fuzzy comprehensive evaluation methods, making it particularly suitable for multi-level evaluation and analysis models that involve difficult-to-quantify indexes. Essentially, AHP-FCE is an improved version of the AHP method that incorporates fuzzy mathematics to handle uncertainties and imprecise data effectively.

The AHP-FCE method involves four fundamental steps: (1) Construction of the Hierarchical Structure, (2) Pairwise Comparisons and Judgmental Matrix, (3) Construction of the membership matrix, and (4) Calculation of Comprehensive Fuzzy Evaluation.

### 3.1. Construct the Hierarchical Structure of the Evaluation Index System

Construct a hierarchical framework system of evaluation indexes based on the screened indexes mentioned above, with the hierarchical structure shown in Figure 3.

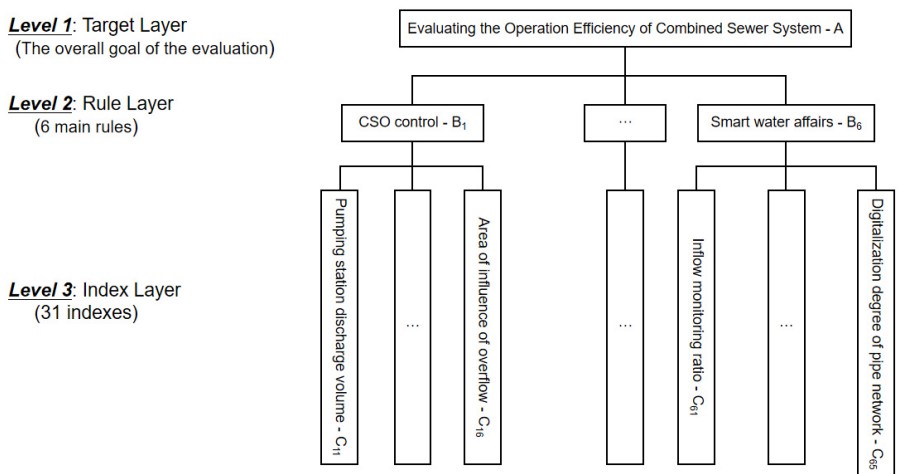

**Figure 3.** Hierarchy tree for the evaluation of operation efficiency in a combined sewer system.

The first level is the Target Layer; its target is to evaluate the operation efficiency of the combined sewer system, which is coded as A.

The second level is the Rule Layer, including the six aspects of "CSO control", "Waterlogging control", "Stable transportation of wastewater", "Management and maintenance of pipeline", "Energy conservation", and "Smart water affairs", which are coded as $B_1$ to $B_6$.

The third level is the Index Layer, which is the specific index corresponding to each rule layer. For example, the six indexes of "pumping station discharge volume", "pumping station discharge frequency", "wastewater treatment plant overflow ratio", "wastewater treatment plant overflow frequency", "overflow potential hazard", and "area of influence of overflow" under the "CSO control" are coded as $C_{11}$ to $C_{16}$; the six indexes of "ponding depth", "ponding frequency", "ponding time ratio", "ponding point ratio", "overloaded pipeline ratio", and "waterlogging reduction potential" under the "Waterlogging control" are coded as $C_{21}$ to $C_{26}$.

### 3.2. Make Pairwise Comparisons and Obtain the Judgmental Matrix

3.2.1. Pairwise Comparisons Scoring

The evaluation indexes under each rule layer were compared in pairs and scored using the 1–9 scale method, following the importance of Saaty elements [28]. The specific scoring method is presented in Table 8 below, and the matrix formed by the scoring results is referred to as the judgment matrix.

**Table 8.** Saaty's nine-point preference scale.

| Intensity of Importance of $a_{ij}$ | Compare Index of i and j |
|---|---|
| 1 | Equally Important |
| 3 | Weakly Important |
| 5 | Strongly Important |
| 7 | Very Strongly Important |
| 9 | Extremely Important |
| 2, 4, 6, 8 | Intermediate value between adjacent |

The elements of the judgment matrix have the following properties [28]:

$$a_{ij} = \frac{1}{a_{ji}} \tag{1}$$

To ensure the reliability and effectiveness of the inquiry results, an expert authority coefficient (Cr) is introduced. The expert authority coefficient (Cr) is determined based on the expert's academic title (C), expert judgment basis (Ca), and expert familiarity (Cs). The scoring method for these factors is presented in Table 9, and the calculation method for the expert authority coefficient (Cr) is shown in Equation (2), as follows:

$$C_r = \frac{1}{C_a + C_s + C} \tag{2}$$

Cr > 0.7 is usually considered acceptable.

**Table 9.** Expert self-assessment authority coefficient (Cr) score sheet.

| Authoritative Coefficient Sub-Items | | Grading Rules | | | | |
|---|---|---|---|---|---|---|
| Basis for judgment (Ca) | Theoretical knowledge | Large 0.3 | Medium 0.2 | Small 0.1 | | |
| | Experience | 0.5 | 0.4 | 0.3 | | |
| | Personal intuition | 0.1 | 0.1 | 0.1 | | |
| | References | 0.1 | 0.1 | 0.05 | | |
| Familiarity (Cs) | | very familiar 1 | familiar 0.8 | general 0.5 | less familiar 0.2 | no familiar 0 |
| Academic title (C) | | Senior 1 | Deputy Senior 0.9 | Medium 0.8 | Primary 0.7 | |

Based on the valid part of the expert survey results, construct a judgment matrix and calculate its eigenvalues; "$\lambda_{max}$" and feature vector "w" are used to empower the weights of various indexes.

3.2.2. Computation of Weights and Consistency of Comparisons

Consistency testing is to check whether there are contradictions between the weights of the indicators obtained by the Analytic Hierarchy Process. The RI values of the 2–7 order judgment matrix are detailed in Table 10.

$$CI = \frac{\lambda_{max} - N}{N - 1} \tag{3}$$

$$CR = \frac{CI}{RI} \tag{4}$$

**Table 10.** RI values comparison table.

| N | 2 | 3 | 4 | 5 | 6 | 7 |
|---|---|---|---|---|---|---|
| RI | 0.00 | 0.58 | 0.90 | 1.12 | 1.24 | 1.32 |

When CR < 0.1 [28], it is considered that the consistency of constructing the judgment matrix is acceptable, that is, the results of index weighting can be used. Equations (3) and (4) come from the literature [28].

### 3.3. Construct the Membership Matrix

This study adopts a two-level fuzzy comprehensive evaluation model.

#### 3.3.1. Determine and Divide the Factor Set

Determine the elements in the factor set U = {$u_1$, $u_2$, ...,$u_3$} corresponding to 31 evaluation indexes [29].

Divide the factor set U = {$u_1$,$u_2$, ...,$u_{31}$} into six groups U = {$U_1$, $U_2$, ..., $U_6$} corresponding to six rule layers, and have the relationship in Equation (5)

$$U = \sum_i^6 U_i \ and \ U_i \bigcap U_j \neq \varnothing (i \neq j) \tag{5}$$

#### 3.3.2. Determine the Comment Set

Determine the comment set V = {$v_1$,$v_2$, ...,$v_m$} [29], according to relevant standards, statistical yearbooks, engineering materials, literature, and policy documents and divide the evaluation level domain into five evaluation levels: A+, A, A−, B, and C, that is, V = {$v_1$, $v_2$, ..., $v_5$}.

The domain of discourse is divided into two types: quantitative indexes and qualitative indexes, as shown in Tables 11 and 12.

**Table 11.** Classification of discourse domain of quantitative evaluation index.

| Code | Index Layer | Level | | | | | Classification Reference |
|------|-------------|-------|---|----|---|---|--------------------------|
| | | A+ | A | A− | B | C | |
| $C_{11}$ | Pumping station discharge volume (m$^3$) | 252 | 266 | 280 | 294 | 308 | Engineering data, Literature [30] |
| $C_{12}$ | Pumping station discharge frequency (%) | 10 | 20 | 30 | 40 | 50 | Engineering data, Water situation bulletin [31] |
| $C_{13}$ | Wastewater treatment plant overflow ratio (%) | 2 | 4 | 6 | 8 | 10 | Engineering data |
| $C_{14}$ | Wastewater treatment plant overflow frequency(%) | 30 | 40 | 50 | 60 | 70 | Engineering data |
| $C_{21}$ | Ponding depth (m) | 0.27 | 0.4 | 0.6 | 0.7 | 1 | Standard [32,33] |
| $C_{22}$ | Ponding frequency (%) | 10 | 15 | 20 | 25 | 30 | Engineering data |
| $C_{24}$ | Ponding point ratio (%) | 4 | 8 | 12 | 16 | 20 | Engineering data |
| $C_{25}$ | Overloaded pipeline ratio (%) | 4 | 8 | 12 | 16 | 20 | Engineering data |
| $C_{31}$ | Variation coefficient of inflow volume (%) | 0.1 | 0.15 | 0.20 | 0.25 | 0.3 | Standard [32] |

**Table 11.** *Cont.*

| Code | Index Layer | Level | | | | | Classification Reference |
|------|-------------|-------|---|---|---|---|--------------------------|
| | | A+ | A | A− | B | C | |
| $C_{34}$ | Forebay water level of outlet pump room (m) | 2.0–2.5 | 1.8 3.2 | 1.5 3.5 | 1.2 3.8 | 0.9 4.0 | Literature [34] |
| $C_{41}$ | Pipeline network length per capita (km/10,000 people) | 28 | 23 | 18 | 13 | 8 | Statistical yearbook [35–40] |
| $C_{42}$ | Urban pipeline network investment (10,000 yuan/km$^2$) | 650 | 500 | 350 | 200 | 50 | Statistical yearbook, Statistical bulletin [41] |
| $C_{43}$ | Wastewater delivery capacity (%) | 90 | 95 | 100 | 110 | 120 | Statistical yearbook |
| $C_{44}$ | Maintenance and update cost (10,000 yuan/km) | 30 | 25 | 20 | 15 | 10 | Statistical yearbook [35–40] |
| $C_{45}$ | Pipeline network density (km/km$^2$) | 19 | 15 | 11 | 7 | 3 | Statistical yearbook [35–40] Literature [42] |
| $C_{46}$ | Wastewater treatment rate (%) | 99.8 | 98.6 | 97.4 | 96.2 | 95 | Statistical yearbook [35–40] |
| $C_{48}$ | Pipeline inspection (%) | 180 | 140 | 100 | 80 | 50 | Policy document [43] |
| $C_{51}$ | Total energy consumption (kW·h) | 7650 | 8075 | 8500 | 8925 | 9350 | Engineering data |
| $C_{52}$ | Wastewater transportation unit consumption (kW·h/1000 m$^3$) | 57.6 | 60.8 | 64 | 67.2 | 70.4 | Engineering data |
| $C_{61}$ | Inflow monitoring ratio (%) | 90 | 70 | 50 | 30 | 10 | Standard [44] |
| $C_{62}$ | Layout of pipeline monitoring points (points/10 km) | 20 | 10 | 4 | 1 | 0.5 | Standard [44] |
| $C_{65}$ | Digitalization degree of pipe network (%) | 90 | 80 | 70 | 60 | 50 | Standard [44] |

**Table 12.** Classification of discourse domain of qualitative evaluation index.

| Code | Index Layer | Level | | | | | Classification Reference |
|------|-------------|-------|---|---|---|---|--------------------------|
| | | A+ | A | A− | B | C | |
| $C_{15}$ | Overflow potential hazard | very small | small | general | large | very large | |
| $C_{16}$ | Area of influence of overflow | very small | small | general | large | very large | |
| $C_{23}$ | Ponding time ratio | very short | short | general | long | very long | |
| $C_{26}$ | Waterlogging reduction potential | very large | large | general | small | very small | |
| $C_{32}$ | Storage capacity | very large | large | general | small | very small | Expert Consultation |
| $C_{33}$ | Pumping station dispatch capability | very good | better | general | poor | worse | |
| $C_{53}$ | Environmental benefits | very good | better | general | poor | worse | |
| $C_{63}$ | Intelligent decision | very good | better | general | poor | worse | |
| $C_{64}$ | Intelligent control | very good | better | general | poor | worse | |

### 3.3.3. Second-Level Factor Set Evaluation Calculation

The fuzzy comprehensive evaluation matrix is obtained by comprehensively evaluating the sets of secondary factors $U_i = \{u_1^{(i)}, u_2^{(i)}, \ldots, u_{ni}^{(i)}\}$ [29], and the membership function is used to calculate the membership degree, as shown in Equation (6):

$$R_i = \begin{bmatrix} r_{11}^{(i)} & r_{12}^{(i)} & \cdots & r_{15}^{(i)} \\ r_{21}^{(i)} & r_{22}^{(i)} & \cdots & r_{25}^{(i)} \\ \vdots & \vdots & \vdots & \vdots \\ r_{n1}^{(i)} & r_{n2}^{(i)} & \cdots & r_{n5}^{(i)} \end{bmatrix} \ (i = 1, 2, \cdots, 6) \tag{6}$$

Through the AHP method, it can be known that the weight of each secondary factor set $U_i = \{u_1^{(i)}, u_2^{(i)}, \ldots, u_{ni}^{(i)}\}$ is $W = \{w_1^{(i)}, w_1^{(i)}, \ldots, w_{ni}^{(i)}\}$ [29], then the comprehensive judgment of each secondary fuzzy judgment matrix is:

$$B_i = W_i \times R_i \ (i = 1, 2, \cdots, 6) \tag{7}$$

In this study, the membership function of semi-trapezoidal fuzzy distribution, which as shown in Figure 4, was used to calculate the membership degree for quantitative indicators, and the formula is as follows [29]:

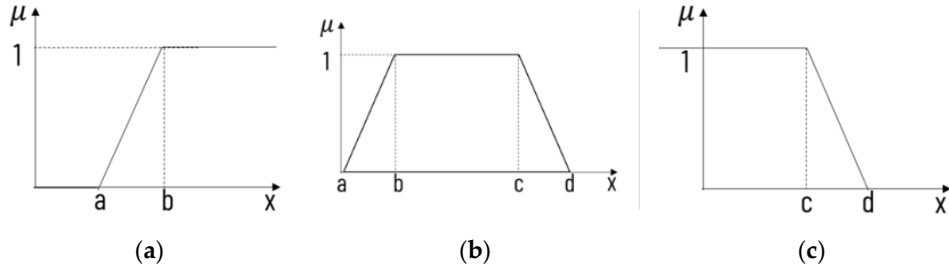

(a)      (b)      (c)

**Figure 4.** Trapezoidal membership function distribution. Panel (**a**) shows the L-function distribution; Panel (**b**) shows the normal distribution; Panel (**c**) shows the R-function distribution.

According to the classification of indexes domains, A+ correspond to Formulas (8), which is L-function (Figure 4a); the three hierarchical domains A, A−, and B correspond to normal function (Figure 4b) Formulas (9); C correspond to Formulas (10), which is R-function (Figure 4c).

$$\mu(x) = \begin{cases} 0, & x < a \\ \frac{x-a}{b-a}, & a \le x \le b \\ 1, & x > b \end{cases} \tag{8}$$

$$\mu(x) = \begin{cases} 0, & x < a \\ \frac{x-a}{b-a}, & a \le x < b \\ 1, & b \le x < c \\ \frac{d-x}{d-c}, & c \le x < d \\ 0, & x \ge b \end{cases} \tag{9}$$

$$\mu(x) = \begin{cases} 1, & x < c \\ \frac{d-x}{d-c}, & c \le x \le d \\ 0, & x > d \end{cases} \tag{10}$$

For qualitative indexes, adopt fuzzy statistical method, invite experts in the industry to evaluate the domain level of qualitative indexes, count the results of expert evaluation, and use the frequency of membership to calculate the degree of membership.

*3.4. Calculation of Comprehensive Fuzzy Evaluation*

Comprehensively evaluate the elements in the first-level factor set U = {U$_1$, U$_2$, ..., U$_6$}. In this study, the six aspects of rule layers of the first-level factors—CSO control, waterlogging control, stable transportation of wastewater, management and maintenance of pipeline, energy conservation, and smart water affairs—are subjectively weighted, and the weights are respectively given to 0.2, 0.2, 0.18, 0.12, 0.15, and 0.15, according to the characteristics of the research object, that is, W = {w$_1$, w$_2$, w$_3$, w$_4$, w$_5$, w$_6$} = {0.2, 0.2, 0.18, 0.12, 0.15, 0.15}, then [29]:

$$R = [B_1, B_2, B_3, B_4, B_5, B_6]^T \tag{11}$$

Comprehensive evaluation is:

$$B = W \times R \tag{12}$$

Finally, the corresponding comments or grades are determined according to the principle of the maximum degree of membership. In order to improve the accuracy of the results, this study adopts the method of the weighted average to determine the degree of membership.

## 4. Case Application

*4.1. Overview of the Situation in a Certain Combined Sewer Area of Shanghai*

Figure 5 illustrates the combined sewer area, which is one of several large areas in the central urban region of Shanghai. This combined sewer system covers an extensive area of 335 square kilometers and serves a substantial population of approximately 6 million people. The drainage network in this area connects with four urban wastewater treatment plants with a combined capacity of about 2.282 million cubic meters per day (m$^3$/d). Specifically, the first wastewater treatment plant has a design capacity of 1.7 million m$^3$/d, while the second wastewater treatment plant has a design capacity of 500,000 m$^3$/d. Following the implementation of upgrades and expansions, the replenishment capacity of the system will be increased to 800,000 m$^3$/d. Consequently, the capacity of the first wastewater treatment plant will be reduced to 1.1 million m$^3$/d, and the capacity of the second wastewater treatment plant will be reduced to 300,000 m$^3$/d. Despite these changes, the overall treated-water volume in the combined sewer area will remain at 2.2 million cubic meters per day to address rainwater overflow issues effectively. Furthermore, the construction of a new wastewater treatment plant with a capacity of 1.2 million m$^3$/d is planned, which will further enhance the operational efficiency of the entire drainage network in the combined sewer area. These improvements and expansions will play a vital role in efficiently managing wastewater and rainwater overflow, contributing to the sustainable development and environmental protection of the urban region.

The study focuses on the evaluation of the combined sewer system in the combined sewer area of Shanghai, encompassing the following components: (1) the trunk line and main branch pipeline network of the combined sewer phase 1 and the combined sewer phase 2; (2) the wastewater treatment plants at the end of the pipeline network; (3) the pumping station for the trunk line and the main branch line of the combined sewer phase 1 and the combined sewer phase 2. Figure 6 provides an overview of the combined sewer area of Shanghai, serving as the geographic representation of the study object.

The research objective is to comprehensively evaluate the operational efficiency of the combined sewer system in the Shanghai combined sewer area over a three-year period (2020, 2021, and 2022) and analyze the results of the evaluation.

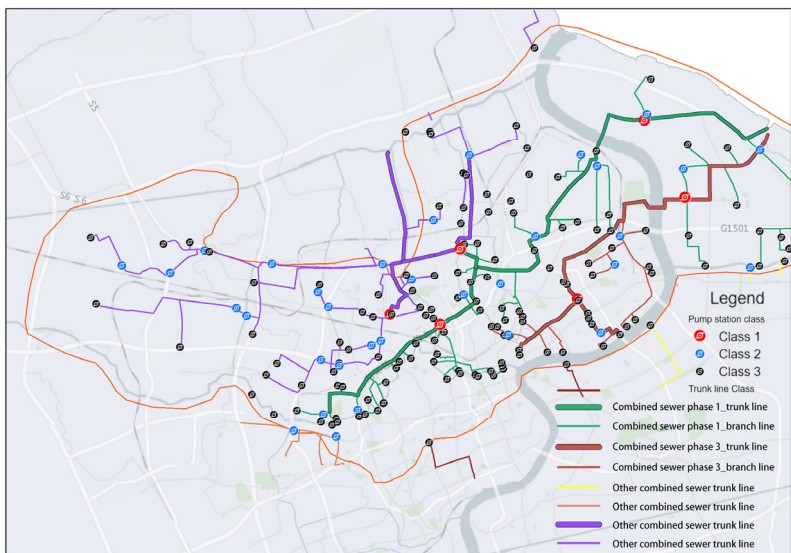

**Figure 5.** The overall map of a certain combined sewer area of Shanghai.

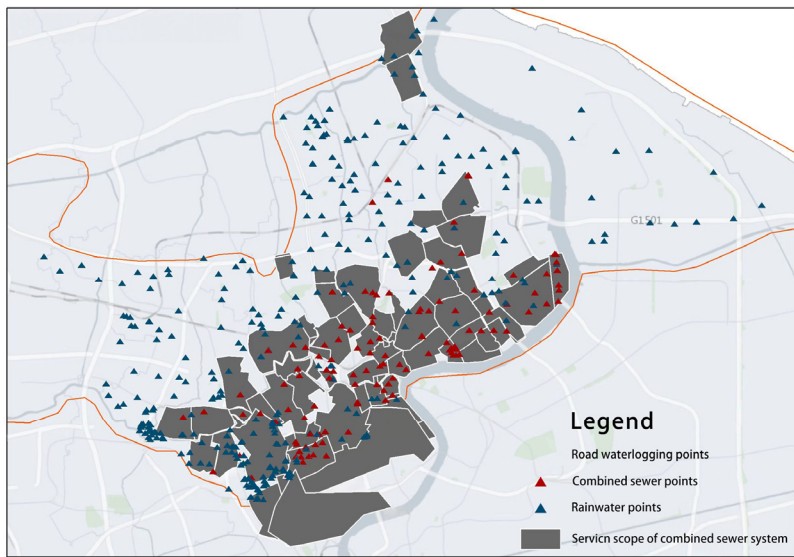

**Figure 6.** Monitoring map of points in a certain combined sewer area of Shanghai.

### *4.2. Evaluating the Operation Efficiency of the Combined Sewer Area*

#### 4.2.1. Empowerment of Indexes Use the AHP Method

In the process of using the Analytic Hierarchy Process (AHP) to assign weights to each index, seven experts specializing in drainage systems were consulted. These experts were asked to compare the indicators under each criterion layer using the 1–9 scale method. Based on their level of authority, the relative importance of each indicator was determined using a weighted average method. Subsequently, a consistency judgment matrix was constructed to calculate the weight of each indicator, and consistency testing was performed to ensure the validity of the results. Taking the indexes in the CSO control of rule layer B1 as an example, the scoring statistics of the importance of Saaty elements for each expert are presented in Table 13. Among the experts, three have an authority coefficient Cr > 0.7 in expert correspondence, with their authority coefficients being 0.8, 0.97, and 0.83, respectively. The pairwise comparison matrix formed by the experts is organized in Table 14. After conducting the consistency test, the calculation results are shown in Table 15.

**Table 13.** Expert scoring statistics of CSO control in the evaluation of combined sewer system.

| Pairwise Comparison Indexes | Expert Scoring Results Combined with Expert Authority Coefficient Weighted Statistics | | | | | | | | | Weighted Average Scaled Results |
|---|---|---|---|---|---|---|---|---|---|---|
| | 1/9 | 1/7 | 1/5 | 1/3 | 1 | 3 | 5 | 7 | 9 | |
| $C_{11}$–$C_{12}$ | 0 | 0 | 0.97 | 0 | 0 | 1.63 | 0 | 0 | 0 | 1.96 |
| $C_{11}$–$C_{13}$ | 0 | 0 | 0 | 0 | 1.8 | 0 | 0.8 | 0 | 0 | 2.23 |
| $C_{11}$–$C_{14}$ | 0 | 0 | 0.97 | 0 | 0 | 0.83 | 0 | 0.8 | 0 | 3.19 |
| $C_{11}$–$C_{15}$ | 0 | 0 | 0 | 0 | 0.97 | 0.8 | 0.83 | 0 | 0 | 2.89 |
| $C_{11}$–$C_{16}$ | 0 | 0 | 0 | 0 | 0.97 | 0 | 1.63 | 0 | 0 | 3.51 |
| $C_{12}$–$C_{13}$ | 0 | 0 | 0.97 | 0.83 | 0 | 0 | 0.8 | 0 | 0 | 1.72 |
| $C_{12}$–$C_{14}$ | 0 | 0 | 0 | 0 | 1.8 | 0.8 | 0 | 0 | 0 | 1.62 |
| $C_{12}$–$C_{15}$ | 0 | 0 | 0 | 0 | 0.97 | 0.83 | 0.8 | 0 | 0 | 2.87 |
| $C_{12}$–$C_{16}$ | 0 | 0 | 0 | 0 | 0.97 | 1.63 | 0 | 0 | 0 | 2.26 |
| $C_{13}$–$C_{14}$ | 0 | 0 | 0.97 | 0 | 0 | 1.63 | 0 | 0 | 0 | 1.96 |
| $C_{13}$–$C_{15}$ | 0 | 0 | 0 | 0 | 0 | 0 | 1.63 | 0.97 | 0 | 5.74 |
| $C_{13}$–$C_{16}$ | 0 | 0 | 0 | 0 | 0 | 0.8 | 0.83 | 0.97 | 0 | 5.13 |
| $C_{14}$–$C_{15}$ | 0 | 0 | 0 | 0 | 1.77 | 0.83 | 0 | 0 | 0 | 1.64 |
| $C_{14}$–$C_{16}$ | 0 | 0 | 0 | 0 | 0.97 | 1.63 | 0 | 0 | 0 | 2.26 |
| $C_{15}$–$C_{16}$ | 0 | 0 | 0 | 0.83 | 1.77 | 0 | 0 | 0 | 0 | 0.79 |

**Table 14.** The pairwise comparison matrix of CSO control in the evaluation of combined sewer system.

| | $C_{11}$ | $C_{12}$ | $C_{13}$ | $C_{14}$ | $C_{15}$ | $C_{16}$ | Eigenvalues (Weights) |
|---|---|---|---|---|---|---|---|
| $C_{11}$ | 1 | 1.96 | 2.23 | 3.19 | 2.89 | 3.51 | 0.3062 |
| $C_{12}$ | 0.51 | 1 | 1.72 | 1.62 | 2.87 | 2.26 | 0.2070 |
| $C_{13}$ | 0.45 | 0.58 | 1 | 1.96 | 5.74 | 5.13 | 0.2259 |
| $C_{14}$ | 0.31 | 0.62 | 0.51 | 1 | 1.64 | 2.26 | 0.1283 |
| $C_{15}$ | 0.35 | 0.35 | 0.17 | 0.61 | 1 | 0.79 | 0.0598 |
| $C_{16}$ | 0.28 | 0.44 | 0.20 | 0.44 | 1.27 | 1 | 0.0698 |
| CR = 0.045 < 0.1 | | | | | | | |

**Table 15.** Evaluation index weight of combined sewer system in a certain combined sewer area of Shanghai.

| Rule Layer | Code | Weight | Index Layer | Code | Local Weight | Global Weight |
|---|---|---|---|---|---|---|
| CSO control | $B_1$ | 0.2 | Pumping station discharge volume | $C_{11}$ | 0.3062 | 0.0612 |
| | | | Pumping station discharge frequency | $C_{12}$ | 0.2070 | 0.0414 |
| | | | Wastewater treatment plant overflow ratio | $C_{13}$ | 0.2259 | 0.0452 |
| | | | Wastewater treatment plant overflow frequency | $C_{14}$ | 0.1283 | 0.0257 |
| | | | Overflow potential hazard | $C_{15}$ | 0.0598 | 0.0120 |
| | | | Area of influence of overflow | $C_{16}$ | 0.0698 | 0.0140 |
| Waterlogging control | $B_2$ | 0.2 | Ponding depth | $C_{21}$ | 0.2961 | 0.0592 |
| | | | Ponding frequency | $C_{22}$ | 0.2151 | 0.0430 |
| | | | Ponding time ratio | $C_{23}$ | 0.0983 | 0.0197 |
| | | | Ponding point ratio | $C_{24}$ | 0.0895 | 0.0179 |
| | | | Overloaded pipeline ratio | $C_{25}$ | 0.2347 | 0.0469 |
| | | | Waterlogging reduction potential | $C_{26}$ | 0.0661 | 0.0132 |
| Stable transportation of wastewater | $B_3$ | 0.18 | Variation coefficient of inflow volume | $C_{31}$ | 0.3599 | 0.0648 |
| | | | Storage capacity | $C_{32}$ | 0.3178 | 0.0572 |
| | | | Pumping station dispatch capability | $C_{33}$ | 0.2411 | 0.0434 |
| | | | Forebay water level of outlet pump room | $C_{34}$ | 0.0813 | 0.0146 |
| Management and maintenance of pipeline | $B_4$ | 0.12 | Pipeline network length per capita | $C_{41}$ | 0.1749 | 0.0210 |
| | | | Urban pipeline network investment | $C_{42}$ | 0.2234 | 0.0268 |
| | | | Wastewater delivery capacity | $C_{43}$ | 0.2256 | 0.0271 |
| | | | Maintenance and update cost | $C_{44}$ | 0.0965 | 0.0116 |
| | | | Pipeline network density | $C_{45}$ | 0.1019 | 0.0122 |
| | | | Wastewater treatment rate | $C_{46}$ | 0.1103 | 0.0132 |
| | | | Pipeline inspection | $C_{47}$ | 0.0674 | 0.0081 |
| Energy conservation | $B_5$ | 0.15 | Total energy consumption | $C_{51}$ | 0.7306 | 0.1096 |
| | | | Wastewater transportation unit consumption | $C_{52}$ | 0.1884 | 0.0283 |
| | | | Environmental benefits | $C_{53}$ | 0.0810 | 0.0122 |
| Smart Water affairs | $B_6$ | 0.15 | Inflow monitoring ratio | $C_{61}$ | 0.3786 | 0.0568 |
| | | | Layout of pipeline monitoring points | $C_{62}$ | 0.2732 | 0.0410 |
| | | | Intelligent decision | $C_{63}$ | 0.1135 | 0.0170 |
| | | | Intelligent control | $C_{64}$ | 0.1189 | 0.0178 |
| | | | Digitalization degree of pipe network | $C_{65}$ | 0.1158 | 0.0174 |

#### 4.2.2. Determining the Degree of Membership Function

In order to handle quantitative indexes, the semi-trapezoidal fuzzy distribution membership function will be applied to calculate the membership degree of each indicator for the years 2020 to 2022. On the other hand, for qualitative indexes, a panel of seven experts in the field of sewer systems will be invited to evaluate the qualitative indicators for the same three-year period (2020 to 2022). The degree of membership for each indicator will be determined using the fuzzy statistical method, taking the evaluation results for the year 2022 as an example, as shown in Table 16.

**Table 16.** Membership degree of the evaluation indexes of operation efficiency in a certain combined sewer area of Shanghai.

| Code | Index Layer | Actual Value | | | 2022 Actual Value of Membership | | | | |
|---|---|---|---|---|---|---|---|---|---|
| | | **2020** | **2021** | **2022** | **A+** | **A** | **A−** | **B** | **C** |
| $C_{11}$ | Pumping station discharge volume (m³) | 292.69 | 325.78 | 223.76 | 1 | 0 | 0 | 0 | 0 |
| $C_{12}$ | Pumping station discharge frequency (%) | 47.40 | 47.12 | 45.75 | 0 | 0 | 0 | 0.425 | 0.575 |
| $C_{13}$ | Wastewater treatment plant overflow ratio (%) | 12.51 | 6.96 | 2.69 | 0.655 | 0.345 | 0 | 0 | 0 |
| $C_{14}$ | Wastewater treatment plant overflow frequency (%) | 91.44 | 77.72 | 38.36 | 0.164 | 0.836 | 0 | 0 | 0 |
| $C_{15}$ | * Overflow potential hazard | — | — | — | 0.286 | 0.429 | 0.286 | 0 | 0 |
| $C_{16}$ | * Area of influence of overflow | — | — | — | 0.143 | 0.571 | 0.286 | 0 | 0 |
| $C_{21}$ | Ponding depth (m) | 1 | 0.98 | 1 | 0 | 0 | 0 | 0 | 1 |
| $C_{22}$ | Ponding frequency (%) | 27.86 | 18.62 | 18.18 | 0 | 0.364 | 0.636 | 0 | 0 |
| $C_{23}$ | *Ponding time ratio | — | — | — | 0.143 | 0.429 | 0.286 | 0.143 | 0 |
| $C_{24}$ | Ponding point ratio (%) | 12.50 | 10.71 | 9.38 | 0 | 0.655 | 0.345 | 0 | 0 |
| $C_{25}$ | Overloaded pipeline ratio (%) | 13.67 | 11.28 | 9.57 | 0 | 0.608 | 0.393 | 0 | 0 |
| $C_{26}$ | * Waterlogging reduction potential | — | — | — | 0.143 | 0.429 | 0.429 | 0 | 0 |
| $C_{31}$ | Variation coefficient of inflow volume (%) | 0.21 | 0.22 | 0.17 | 0 | 0.6 | 0.4 | 0 | 0 |
| $C_{32}$ | Storage capacity | — | — | — | 0.286 | 0.429 | 0.286 | 0 | 0 |
| $C_{33}$ | Pumping station dispatch capability | — | — | — | 0.143 | 0.286 | 0.571 | 0 | 0 |
| $C_{34}$ | Forebay water level of outlet pump room (m) | 2.64 | 2.56 | 2.50 | 1 | 0 | 0 | 0 | 0 |
| $C_{41}$ | Pipeline network length per capita (km/10,000 people) | 11.63 | 11.68 | 11.62 | 0 | 0 | 0 | 0.724 | 0.276 |
| $C_{42}$ | Urban pipeline network investment (10,000 yuan/km²) | 530.59 | 577.94 | 532.28 | 0.215 | 0.785 | 0 | 0 | 0 |
| $C_{43}$ | Wastewater delivery capacity (%) | 119.12 | 111.19 | 100.44 | 0 | 0 | 0.956 | 0.044 | 0 |
| $C_{44}$ | Maintenance and update cost (10,000 yuan/km) | 17.57 | 13.39 | 24.20 | 0 | 0.84 | 0.16 | 0 | 0 |
| $C_{45}$ | Pipeline network density (km/km²) | 13.92 | 19.08 | 20.03 | 1 | 0 | 0 | 0 | 0 |
| $C_{46}$ | Wastewater treatment rate (%) | 96.68 | 96.89 | 97.38 | 0 | 0 | 0.983 | 0.017 | 0 |
| $C_{47}$ | Pipeline inspection (%) | 120 | 118 | 102 | 0 | 0.05 | 0.95 | 0 | 0 |
| $C_{51}$ | Total energy consumption (kW·h) | 9217.87 | 8455.60 | 8096.07 | 0 | 0.950 | 0.050 | 0 | 0 |
| $C_{52}$ | Wastewater transportation unit consumption (kW·h/1000 m³) | 69.77 | 61.13 | 59.72 | 0.338 | 0.663 | 0 | 0 | 0 |
| $C_{53}$ | Environmental benefits | — | — | — | 0.571 | 0.143 | 0.286 | 0 | 0 |
| $C_{61}$ | Inflow monitoring ratio (%) | 50 | 50 | 50 | 0 | 0 | 1 | 0 | 0 |
| $C_{62}$ | Layout of pipeline monitoring points (points/10 km) | 20 | 20 | 20 | 1 | 0 | 0 | 0 | 0 |
| $C_{63}$ | Intelligent decision | — | — | — | 0 | 0.143 | 0.429 | 0.286 | 0.143 |
| $C_{64}$ | Intelligent control | — | — | — | 0.143 | 0.429 | 0.143 | 0.286 | 0 |
| $C_{65}$ | Digitalization degree of pipe network (%) | 100 | 100 | 100 | 1 | 0 | 0 | 0 | 0 |

Note: The indexes marked with * are qualitative indexes, and their actual values are scored by experts.

4.2.3. Multi-Level Fuzzy Comprehensive Evaluation and Quantitative Scoring

According to formulas (6) and (7), the fuzzy judgment matrix is constructed and calculated for the six aspects of the rule layer for the three years 2020 to 2022. The evaluation results for the rule layer are presented in Table 17.

**Table 17.** Evaluation results and quantitative score of combined sewer system operational efficiency rule layer in a certain combined sewer area of Shanghai from 2020 to 2022.

| Years | Rule Layer | Code | Evaluation Level Proportion % | | | | | Quantitative Score |
| | | | A+ | A | A− | B | C | |
|---|---|---|---|---|---|---|---|---|
| 2020 | CSO control | $B_1$ | 1 | 2.71 | 10.41 | 34.85 | 50.74 | 66.61 |
| | Waterlogging control | $B_2$ | 3.29 | 8.45 | 26.20 | 20.12 | 41.91 | 71.09 |
| | Stable transportation of wastewater | $B_3$ | 14.49 | 26.67 | 48.20 | 10.64 | 0 | 84.50 |
| | Management and maintenance of pipeline | $B_4$ | 4.56 | 28.59 | 15.49 | 25.99 | 25.37 | 76.10 |
| | Energy conservation | $B_5$ | 4.63 | 2.31 | 1.16 | 26.42 | 65.48 | 65.42 |
| | Smart water affairs | $B_6$ | 43.92 | 8.34 | 46.12 | 1.62 | 0 | 89.42 |
| 2021 | CSO control | $B_1$ | 0 | 3.70 | 18.30 | 19.51 | 58.19 | 66.54 |
| | Waterlogging control | $B_2$ | 2.35 | 21.50 | 45.58 | 2.92 | 27.64 | 76.79 |
| | Stable transportation of wastewater | $B_3$ | 15.42 | 17.76 | 48.99 | 17.84 | 0 | 83.08 |
| | Management and maintenance of pipeline | $B_4$ | 21.80 | 13.77 | 10.05 | 43.98 | 10.41 | 79.27 |
| | Energy conservation | $B_5$ | 3.47 | 26.84 | 69.68 | 0 | 0 | 83.37 |
| | Smart water affairs | $B_6$ | 43.92 | 6.72 | 46.12 | 3.24 | 0 | 89.13 |
| 2022 | CSO control | $B_1$ | 50.23 | 25.07 | 3.70 | 8.80 | 11.90 | 89.05 |
| | Waterlogging control | $B_2$ | 2.35 | 35.00 | 31.62 | 1.40 | 29.61 | 77.89 |
| | Stable transportation of wastewater | $B_3$ | 20.65 | 42.10 | 37.25 | 0 | 0 | 88.34 |
| | Management and maintenance of pipeline | $B_4$ | 15.00 | 25.98 | 40.36 | 13.84 | 4.83 | 83.26 |
| | Energy conservation | $B_5$ | 10.99 | 83.08 | 5.94 | 0 | 0 | 90.51 |
| | Smart water affairs | $B_6$ | 40.60 | 6.72 | 44.42 | 6.64 | 1.62 | 87.80 |

Next, the fuzzy membership degree matrix of the target layer is constructed according to Formulas (11) and (12). Finally, the evaluation results for the three years 2020 to 2022 are calculated and presented in Table 18.

**Table 18.** The final results evaluation and quantified scores of the operation efficiency of the sewer system in the combined sewer area of Shanghai from 2020 to 2022.

| Years | Evaluation Level Proportion % | | | | | Result | Score |
| | A+ | A | A− | B | C | | |
|---|---|---|---|---|---|---|---|
| 2020 | 11.29 | 12.06 | 24.95 | 20.24 | 31.40 | B | 75.15 |
| 2021 | 12.97 | 14.92 | 40.17 | 13.46 | 18.41 | A− | 79 |
| 2022 | 23.77 | 36.18 | 26.17 | 4.70 | 9.12 | A | 86 |

To provide a more direct representation of the evaluation of various indexes of combined sewer operation efficiency, quantitative scoring is performed for each indicator using the degree of membership obtained from the criterion layer. The quantification method involves assigning grades A+, A, A−, B, and C, with corresponding scores of 100, 90, 80, 70, and 60 points, respectively. The sum of these scores, multiplied by the corresponding evaluation grade proportion, yields the final quantitative score, as shown in Tables 17 and 18.

### 4.3. Analysis of Operation Efficiency Evaluation Results

According to Figures 7 and 8, the ranking of the relative importance of the index layer relative to the target layer is Total energy consumption ($C_{51}$) > Variation coefficient of inflow volume ($C_{31}$) > Pumping station discharge volume ($C_{11}$) > Ponding depth ($C_{21}$) > Storage capacity ($C_{32}$) > Inflow monitoring ratio ($C_{61}$) > Overloaded pipeline ratio ($C_{25}$) > Wastewater treatment plant overflow ratio ($C_{13}$) > Pumping station dispatch capability ($C_{33}$) > Ponding frequency ($C_{22}$) > Pumping station discharge frequency ($C_{12}$) > Layout of pipeline monitoring points ($C_{62}$) > Wastewater transportation unit consumption ($C_{52}$) > Wastewater delivery capacity ($C_{43}$) > Urban pipeline network investment ($C_{42}$) > Wastewater treatment plant overflow frequency ($C_{14}$) > Pipeline network length per capita ($C_{41}$) > Ponding time ratio ($C_{23}$) > Ponding point ratio ($C_{24}$) > Intelligent control ($C_{64}$) > Digitalization degree of pipe network ($C_{65}$) > Intelligent decision ($C_{63}$) > Forebay water level of outlet pump room ($C_{34}$) > Area of influence of overflow ($C_{16}$) > Wastewater treatment rate ($C_{46}$) > Waterlogging reduction potential ($C_{26}$) > Pipeline network density ($C_{45}$) > Environmental benefits ($C_{53}$) > Overflow potential hazard ($C_{15}$) > Maintenance and update cost ($C_{44}$) > Pipeline inspection ($C_{47}$).

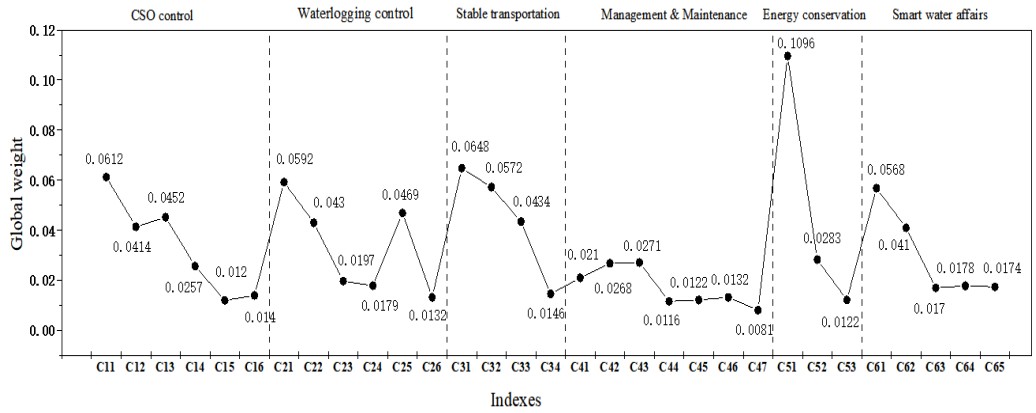

**Figure 7.** Distribution of the global weight of 31 indexes in index layer by AHP.

Note that there are 12 indexes with a total weight ratio of more than 0.04, including Total energy consumption ($C_{51}$), Variation coefficient of inflow volume ($C_{31}$), Pumping station discharge volume ($C_{11}$), Ponding depth ($C_{21}$), Storage capacity ($C_{32}$), Inflow monitoring ratio ($C_{61}$), Overloaded pipeline ratio ($C_{25}$), Wastewater treatment plant overflow ratio ($C_{13}$), Pumping station dispatch capability ($C_{33}$), Ponding frequency ($C_{22}$), Pumping station discharge frequency ($C_{12}$), and Layout of pipeline monitoring points ($C_{62}$); the sum of the weights of these indexes is 0.6697, which is an important index affecting the operation efficiency of the combined sewer system.

It can be seen from Table 17 that in 2020, the CSO control ($B_1$) of the combined sewer system in the combined sewer area of Shanghai can be judged as C, the Waterlogging control ($B_2$) can be judged as B, the Stable transportation of wastewater ($B_3$) can be judged as A−, the Management and maintenance of pipeline ($B_4$) can be judged as A−, the Energy saving ($B_5$) can be judged as C, and the Smart water level ($B_6$) can be judged as A. In 2021, the CSO control ($B_1$) of the combined sewer system in the combined sewer area of Shanghai can be judged as C, the Waterlogging control ($B_2$) can be judged as A−, the Stable transportation of wastewater ($B_3$) can be judged as A−, the Management and maintenance of pipeline ($B_4$) can be judged as A−, the Energy saving ($B_5$) can be judged as A−, and the Smart water level ($B_6$) can be judged as A. In 2021, the CSO control ($B_1$) of the combined sewer system in the combined sewer area of Shanghai can be judged as A, the Waterlogging control ($B_2$) can be judged as A−, the Stable transportation of wastewater ($B_3$) can be judged as A, the Management and maintenance of pipeline ($B_4$) can be judged as A−, the Energy saving ($B_5$) can be judged as A, and the Smart water level ($B_6$) can be judged as A.

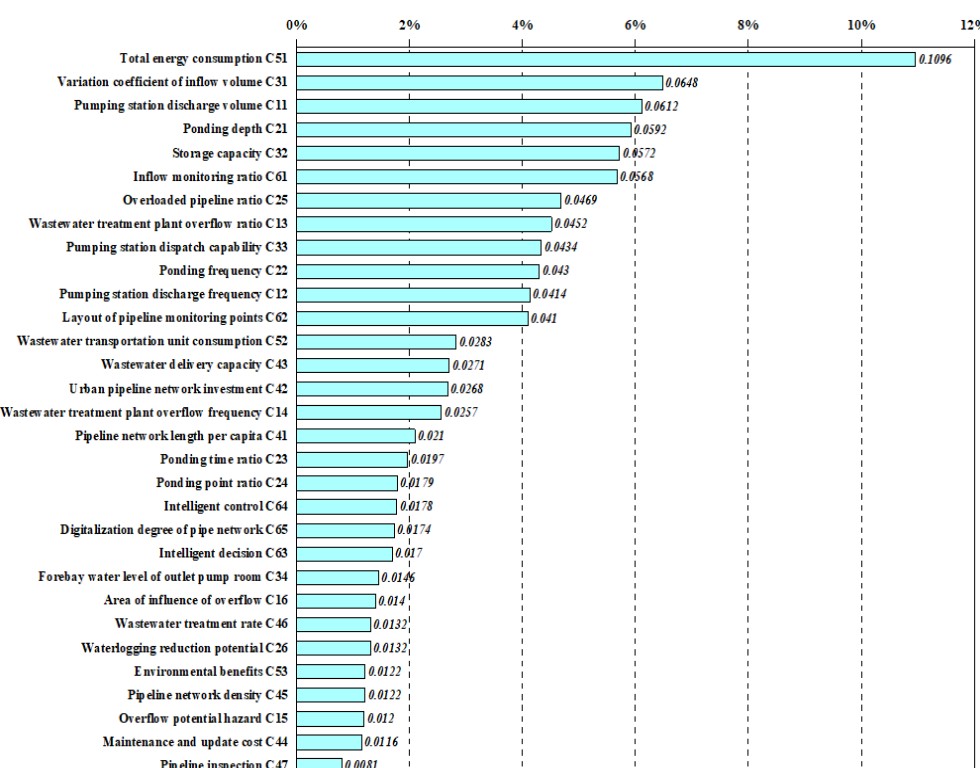

**Figure 8.** Overall prioritization of all 31 indexes by the AHP.

　　The comprehensive evaluation of the operation efficiency of the combined sewer area of Shanghai for the three years from 2020 to 2022 is shown in Table 15. In 2020, the comprehensive evaluation result has a probability of A+ of 11.29%, a probability of A of 12.06%, a probability of A− of 24.95%, a probability of B of 20.24%, and a probability of C of 31.04%. In 2021, the comprehensive evaluation result has a probability of 12.97% at A+, 14.92% at A, 40.17% at A−, 13.46% at B, and 18.41% at C. In 2022, the comprehensive evaluation result has a probability of 23.77% at A+, 36.18% at A, 26.17% at A−, 4.70% at B, and 9.12% at C. In summary, it can be considered that the operation efficiency of the combined sewer system in the combined sewer area will be at a relatively low level in 2020, will increase to a moderate level in 2021, and will further increase to a good level in 2022.

　　Based on the three-year comprehensive evaluation results, it can be concluded that the operational efficiency of the combined sewer system in the Shanghai combined sewer area has been continuously improving. Figure 9 illustrates that this improvement is mainly attributed to the significant advancements in CSO control and energy conservation within the combined sewer system. The consistent enhancement of integrated overflow control in the area can be attributed to the upgrading and replenishment project of wastewater treatment plants in Shanghai, which has effectively curtailed overflow flow and frequency, while reducing the scope of overflow pollution.

　　Furthermore, the continuous improvement in energy conservation within the combined sewer system indicates substantial progress in enhancing quality and efficiency. With decreasing wastewater transportation consumption and total energy consumption, environmental benefits have also been on the rise.

　　However, relatively low scores in Waterlogging control and Management and maintenance of pipeline aspects highlight areas that require more attention and improvement in the future. These aspects represent the main directions for future enhancement in the combined sewer area. Strengthening drainage and waterlogging prevention infrastructure and increasing investment in pipeline maintenance are crucial steps in further improving the operational efficiency of the combined sewer system in the area. Addressing these aspects will help in achieving even higher levels of efficiency and environmental protection.

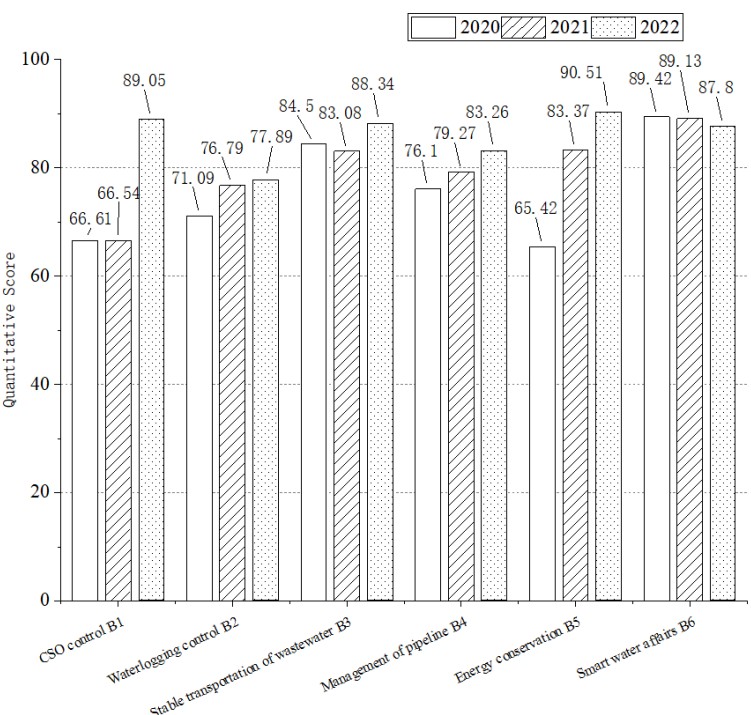

**Figure 9.** The 2020–2022 histogram of quantified score at the rule layer in combined sewer area of Shanghai.

Figures 10 and 11 depict statistical maps illustrating changes in quantitative scores for the index layer between 2020 and 2022. Notably, several indicators, including Pumping station discharge volume (C11), Wastewater treatment plant overflow ratio (C13), Wastewater treatment plant overflow frequency (C14), Ponding frequency (C22), Overloaded pipeline ratio (C25), Wastewater delivery capacity (C43), Maintenance and update cost (C44), Pipeline network density (C45), Total energy consumption (C51), and Wastewater transportation unit consumption (C52), exhibit significant score improvements during this period.

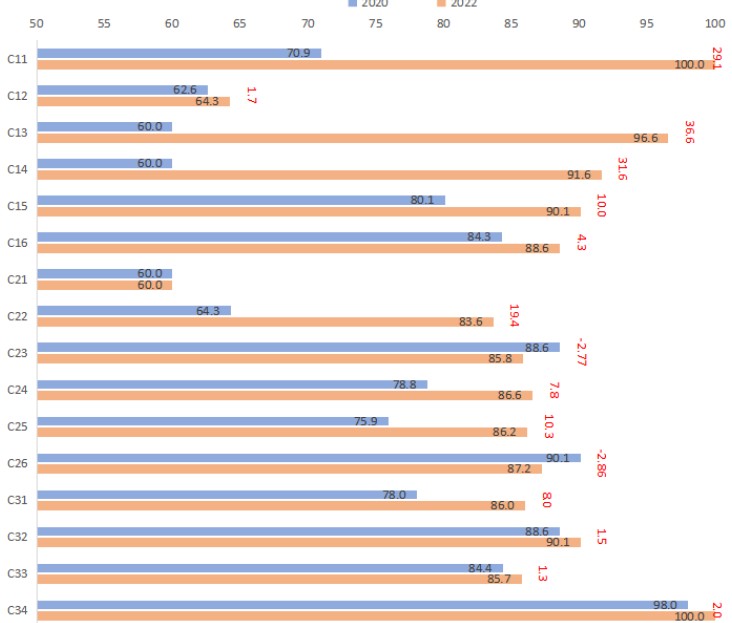

**Figure 10.** Changes in Indicator Tier Scores from 2020 to 2022, Part 1.

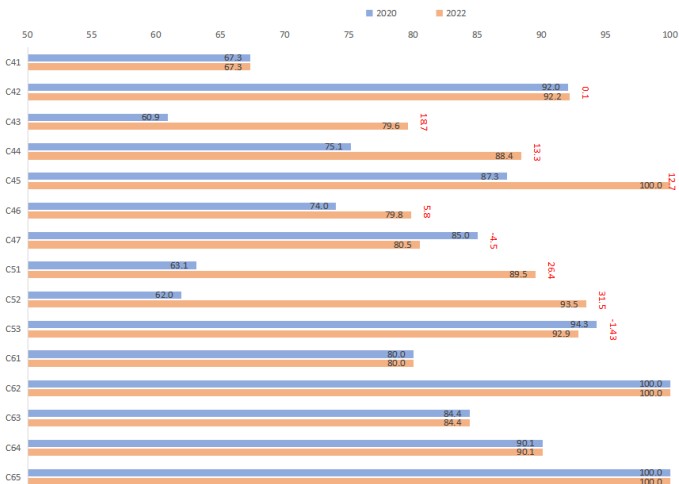

**Figure 11.** Changes in Indicator Tier Scores from 2020 to 2022, Part 2.

Conversely, Pumping station discharge frequency (C12), Wastewater treatment plant overflow ratio (C13), Ponding depth (C21), Pipeline network length per capita (C41), Inflow monitoring ratio (C61), and Intelligent decision (C63) indicators, which previously had lower scores, represent key focus areas for future development and enhancement of the drainage system.

According to the analysis and discussion of the above results, it can be seen that applying this evaluation system can give us a preliminary understanding of the aspects for the improvement of comprehensive operational efficiency and what are the more important indicators that affect the improvement of comprehensive efficiency. We will conduct in-depth research on the contribution, correlation, sensitivity, and other issues between upper- and lower-level indicators, as well as those under the same rule layer, which is the foundation to find applicable theories and algorithms, achieve multi-criteria decision analysis, and even explore the possibility of interdisciplinary cooperation [45].

## 5. Conclusions

The previous studies mainly focus on the evaluation of a specific objective of the sewer system, such as overflow control evaluation, safety risk evaluation, performance evaluation, and so on. A set of comprehensive evaluation systems for assessing the operation efficiency of an urban combined sewer system based on AHP-FCE is established, which originally introduces the evaluation indexes and their evaluation rules related to the level of smart water affairs and the stable transportation of drainage. This study arrived at the following conclusions:

(1) Multiple rounds of screening were conducted using methods such as the coefficient of variation to identify specific indexes that can accurately characterize the operation efficiency of the combined sewer system. Ultimately, a total of 31 specific indexes were identified, covering six essential aspects: CSO control, Waterlogging control, Stable transportation of wastewater, Management and maintenance of pipeline, Energy conservation, and Smart water affairs. This comprehensive evaluation index system model was constructed to assess the operation efficiency of urban combined sewer systems.

(2) The constructed operation efficiency model for the combined sewer system was then applied to the combined sewer area of Shanghai, enabling a thorough evaluation of its operational efficiency over the period from 2020 to 2022. The results highlighted 12 critical influencing indexes in the combined sewer system in this area: total energy consumption, variation coefficient of inflow volume, pumping station discharge volume, ponding depth, storage capacity, inflow monitoring ratio, overloaded pipeline ratio, wastewater treatment plant overflow ratio, pumping station dispatch capability,

ponding frequency, pumping station discharge frequency, and layout of pipeline monitoring points.

(3) The comprehensive evaluation results for the three-year period indicate continuous improvement in the combined sewer system's operational efficiency in this area. The system has progressed from a relatively low level to a relatively good level. Notably, there have been significant enhancements in CSO control and energy conservation. However, there is still scope for improvement in waterlogging control and the management and maintenance of pipelines, highlighting these aspects as primary areas of focus for further enhancement in the future.

**Author Contributions:** Conceptualization, H.W. and M.Y.; methodology, H.W. and M.Y.; software, M.Y.; validation, Y.G., Y.W. and X.D.; formal analysis, M.Y.; investigation, Wang Hongwu and M.Y.; resources, H.W.; data curation, M.Y.; writing—original draft preparation, H.W. and M.Y.; writing—review and editing, H.W. and M.Y.; visualization, M.Y.; supervision, Y.G.; project administration, X.D.; funding acquisition, H.W. and X.D. All authors have read and agreed to the published version of the manuscript.

**Funding:** This research was funded by the National Key Research and Development Program of China: No. 2021YFC3201504 and No. 2021YFC3201305; the Project of Science and Technology Commission of Shanghai Municipality: 20dz1204605.

**Data Availability Statement:** Data cannot be shared publicly, because data from this study contain information about the national security and geography.

**Conflicts of Interest:** The authors declare no conflict of interest.

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
