# Peer review of "An Evaluation System for Assessing the Operational Efficiency of Urban Combined Sewer Systems Using AHP—Fuzzy Comprehensive Evaluation: A Case Study in Shanghai, China"

_water, doi:10.3390/w15193434_

Round 1

Reviewer 1 Report

In this manuscript, authors established a comprehensive evaluation system to evaluate the combined sewer system's operation efficiency. Besides, authors applied the system to assess the operation efficiency of a specific area in Shanghai from 2020 to 2022. Reviewer think that the research shows high practical application value. However, some details should be improved. Thus, I have the following concerns which might require further revision before the manuscript can be accepted.

1. Key words: Punctuations need to be checked carefully.

2. Introduction: The introduction was confusing. The introduction section is an extension of the background description in the abstract section. Authors spent too much text emphasizing the necessity of rainwater runoff management. It was redundant. Please shorten the first paragraph of introduction section. Besides, the summary of evaluation methods was lack of logic. This section should summary of the literature rather listed the literature simply. Introduction should summary the literature to obtain the aim of this research. Thus, it is necessary to adjust the structure of the introduction and rewritten the introduction section.

3. In section 2.2, it is better to add relevant references, such as the indexes’ selection.

4. Please normalize the figures. For example, the Figure.1 should be numbered as Fig.1a, b, c. Please provides clearer Figures.4 and 5, the graphic illustration is fuzzy.

5. Conclusions: At the beginning of conclusions, it is better to add one sentence to summarize the manuscript. 

Reviewer 2 Report

The authors developed a comprehensive evalutaion system through fuzzy hierarchical analysis for evaluating the operation efficiency of combined sewer system. For the screening of the evaluation indexes, this study adopted the coefficient of variation method coupled with the expert scoring method. In addition, the authors applied the evaluation system to assess the operation efficiency of a combined drainage area in Shanghai from 2020 to 2022. The study has possibale practical application value. However, I have the following suggestions and the manuscript need a minor revision.

1. Keywords: Incorrect formatting has occurred, and needs to be changed to the standardized same format. The current keywords are insufficient to fully cover the content of the research, and the authors may add another one or two keywords.

2. Introduction: The content is lengthy. The literature review section needs to be further refined, in which the representative scientific research results and cutting-edge research in the relevant study should be summarized clearly.

3. Section 2: It is suggested to attach the sources or references of the preliminary selection of evaluation indexes to increase the reliability of the preliminary selection of indicators.

4. Figures 4 and 5 in Section 3 have the following problems: Firstly, the figures lack explanatory notes; Secondly, the quality of the figures is not high enough; Finally, the legends are not clear enough to fully show the information.

5. Section 4: The discussion of practical application cases is not in-depth enough. It is better to further discuss the changes in the scores of each evaluation index.

Reviewer 3 Report

The manuscript concerns the issue of evaluation system for assessing the operational efficiency of urban combined sewer systems using fuzzy analytical hierarchy process: a case study in Shanghai, China. In order to adequately assess the operational effectiveness of the combined sewer system, a comprehensive evaluation framework was devised. This framework amalgamated expert judgment and the coefficient of variation technique, encompassing a total of 31 specific metrics. These metrics were utilized to evaluate six fundamental aspects: management of combined sewer overflow (CSO), mitigation of waterlogging, reliable conveyance of wastewater, upkeep and administration of pipelines, energy efficiency, and integration of intelligent water management practices. The evaluation process involved the utilization of the Fuzzy Analytic Hierarchy Process (FAHP), which integrated Analytic Hierarchy Process (AHP) principles for assigning weightage to indices, and fuzzy comprehensive evaluation for quantifiable scoring. The application of this system was demonstrated through the assessment of the operational efficiency within a particular area of Shanghai spanning from 2020 to 2022. Remarks and comments: The fundamental explanation of the Analytic Hierarchy Process (AHP) should be relocated to the appendix section. Additionally, when referring to the source, it would be beneficial to enhance the reference in relation to the classification grounded in the unique characteristics of each factor. It's crucial to highlight that factors within a specific tier are interconnected, as they are reliant on higher-tier factors or hold influence over them. Simultaneously, they exert control over lower-tier factors while being impacted by them, eg. Most Searched Topics in the Scientific Literature on Failures in Photovoltaic Installations. Energies 2022, 15, 8108. https://doi.org/10.3390/en15218108. On what base the index layers were chosen? What motivated the selection of these specific methods for the analysis? This rationale should be emphasized within the overarching structure of the presented study. Why is this particular approach deemed the most suitable solution for conducting the analysis? The integration of expert knowledge is a significant aspect of the research. I believe additional elaboration is necessary on this matter, including which experts were engaged and whether any disparities or contradictions were identified in their perspectives. It's essential to provide references for all equations, particularly if you're not the author of those equations. The potential implications and capabilities of the proposed approach could also be elucidated.

Round 2

Reviewer 3 Report

It is crucial to emphasize the distinctive contributions and originality of this paper. It is important to assess the additional value of the research in order to highlight its significance and uniqueness within the field. Regarding future work, there are various perspectives that can be taken into account. One potential direction for future research could involve conducting empirical studies to verify the effectiveness and practicality of the proposed approach in real-world situations. Furthermore, it would be valuable to examine how the approach can be scaled and adapted to different contexts, gaining valuable insights. Additionally, it is worth exploring potential synergies with related research areas and exploring interdisciplinary collaborations to expand the scope and impact of this work.

It is crucial to emphasize the distinctive contributions and originality of this paper. It is important to assess the additional value of the research in order to highlight its significance and uniqueness within the field. Regarding future work, there are various perspectives that can be taken into account. One potential direction for future research could involve conducting empirical studies to verify the effectiveness and practicality of the proposed approach in real-world situations. Furthermore, it would be valuable to examine how the approach can be scaled and adapted to different contexts, gaining valuable insights. Additionally, it is worth exploring potential synergies with related research areas and exploring interdisciplinary collaborations to expand the scope and impact of this work.

Author Response

Reviewer #3: It is crucial to emphasize the distinctive contributions and originality of this paper. It is important to assess the additional value of the research in order to highlight its significance and uniqueness within the field. Regarding future work, there are various perspectives that can be taken into account. One potential direction for future research could involve conducting empirical studies to verify the effectiveness and practicality of the proposed approach in real-world situations. Furthermore, it would be valuable to examine how the approach can be scaled and adapted to different contexts, gaining valuable insights. Additionally, it is worth exploring potential synergies with related research areas and exploring interdisciplinary collaborations to expand the scope and impact of this work.

Response: Thank you very much for taking the time to review our paper again and also for your valuable suggestions for improvement. We revised our paper based on your comments to accomplish further improvements in the quality of our paper. We have included further research needed in the future and a description of the contributions and originality of this work at the end of section 4 (Line 496-503) and the beginning of part 5 (Line 505-510). A literature numbered 45 has been added.

Line 496-503 :  “According to the analysis and discussion of the above results, it can be seen that applying this evaluation system can give us a preliminary understanding of the aspects for the improvement of comprehensive operational efficiency and what are the more important indicators that affect the improvement of comprehensive efficiency. We will conduct in-depth research on the contribution, correlation, sensitivity, and other issues between upper and lower level indicators, as well as those under the same rule layer, which is the foundation to find applicable theories and algorithms, achieve multi criteria decision analysis, and even explore the possibility of interdisciplinary cooperation [45]. ”

Line 505-510:  “The previous studies mainly focus on the evaluation of a specific objective of the sewer system, such as overflow control evaluation, safety risk evaluation, performance evaluation and so on. A set of comprehensive evaluation system for assessing the operation efficiency of urban combined sewer system based on AHP-FCE is established, which is originally introduces the evaluation indexes and their evaluation rules related to the level of smart water affair and the stable transportation of drainage.”

  1. Kut, P.; Pietrucha-Urbanik, K. Most Searched Topics in the Scientific Literature on Failures in Photovoltaic Installations. Energies. 2022, 15(21).